# Facile Synthesis of Microporous Ferrocenyl Polymers Photocatalyst for Degradation of Cationic Dye

**DOI:** 10.3390/polym14091900

**Published:** 2022-05-06

**Authors:** Bing Zhang, Zhiqiang Tan, Yinhu Zhang, Qingquan Liu, Qianxia Li, Gen Li

**Affiliations:** Hunan Provincial Key Laboratory of Advanced Materials for New Energy Storage and Conversion, School of Materials Science and Engineering, Hunan University of Science and Technology, Taoyuan Street, Xiangtan 411201, China; bingz@mail.hnust.edu.cn (B.Z.); zhiqiangtan@mail.hnust.edu.cn (Z.T.); yhzhang@mail.hnust.edu.cn (Y.Z.); qxli@mail.hnust.edu.cn (Q.L.)

**Keywords:** microporous polymers, ferrocenyl, photocatalyst, cationic dye

## Abstract

Microporous organic polymers (MOPs) were prepared by condensation reactions from substituent-group-free carbazole and pyrrole with 1,1′-ferrocenedicarboxaldehyde without adding any catalysts. The resultant MOPs were insoluble in common solvent and characterized by FTIR, XPS, TGA and SEM. An N_2_ adsorption test showed that the obtained polymers PFcMOP and CFcMOP exhibited Brunauer–Emmett–Teller (BET) surface areas of 48 and 105 m^2^ g^−1^, respectively, and both polymers possessed abundant micropores. The MOPs with a nitrogen and ferrocene unit could be potentially applied in degrading dye with high efficiency.

## 1. Introduction

Microporous organic polymers [1] consisting of light elements such as C, H, N, O etc., featuring tunable functionality and exceptionally high BET surface area, have received considerable academic and industrial attention due to their promising applications in the field of heterogeneous catalysis [2,3], chemical sensing [4,5], selective gas adsorption [6], dye adsorption [7], light harvesting [8], drug delivery [9], electronic or photoluminescence devices, and so on over the past decades. A large number of purely organic polymers with micro or mesopores have been prepared, e.g., hypercrosslinked polymers (HCPs) [10,11,12], polymers of intrinsic microporosity (PIMs) [13], conjugated microporous polymers (CMPs) [14,15], porous aromatic frameworks (PAFs) [16] and covalent organic frameworks (COFs) [17]. Most of the MOPs possessed high BET surface areas and permanent porosity and showed features of high chemical and hydrothermal stability and low density. In addition, it eased the preparation with abundant candidate monomers and mature synthesis technology and the related chemical reactions including Yamamoto couplings [18], Ullmann couplings [19], Suzuki couplings [20], Sonogashira–Hagihara couplings [21], imide formation reactions [22] and “click” chemistry [23] as well as various condensation polymerizations [24,25,26,27,28].

Dye molecules, which have been extensively utilized in various industries, have caused tremendous environmental pollution due to the poor biodegradation of organic dyes [29]. Therefore, the treatment of dye wastewater is of great importance for the environmental protection. To this end, a large number of catalysts, such as hydrogenated graphitic carbon nitride [30], TiO_2_ [31], AC-TiO_2_ [32], CaBi_2_O_4_ [33], CaCO_3_/Ag_2_CO_3_/AgI/Ag composites [34], Ag@AgCl [35], Cu@MnO_2_ nanowires [36], etc., have been reported to decompose organic dyes. In recent years, MOPs with light-responsive catalytic properties have attracted numerous interest [37,38]. Yu et al. have reported serials of microporous polymers photocatalyst (CMP-CSUs) via the oxidative coupling polymerization approach. They reported that CMP-CSUs can be used to catalyze C-3 formylation and the thiocyanation of indoles reactions and Ugi-type reaction [39,40]. Although significant progress toward photoactive MOPs with light-harvesting and catalytic properties has already been achieved, few research studies were available on the direct use of MOPs for photocatalytically degrading organic dye contaminant.

In this work, two hypercrosslinked polymer networks were synthesized by facile catalyst-free polycondensation from substituent-group-free carbazole and pyrrole with 1,1′-ferrocenedicarboxaldehyde, using 1,4-dioxane as a reaction medium. The abundance of carbazole, pyrrole and iron atoms in the polymeric skeleton enables us to examine the photoactive catalysis property of the polymers toward dye molecules, and the cationic methylene blue (MB) was used as a model dye. Notably, the as-prepared MOPs show excellent chemical stability, and they could be used to degrade dye with high efficiency.

## 2. Results and Discussion

As shown in Figure 1, the nitrogen-rich and ferrocene-based microporous polymers networks were synthesized in 1,4-dioxane via one-step polycondensation at 220 °C from substituent-group-free carbazole and pyrrole with 1,1′-ferrocenedi-carboxaldehyde without adding any catalyst. The resultant solid polymers showed complete insolubility in any organic solvent, for example, tetrahydrofuran (THF), dimethyl sulfoxide (DMSO), and N,N-dimethylformamide (DMF), suggesting the hypercrosslinked nature of polymers’ structures. The thermal stability was tested through the thermogravimetric analysis (TGA) under inert atmosphere at temperature from 25 to 800 °C using a heating rate of 10 °C/min (Appendix A, ESI†). A small loss of polymers weight was detected in the preliminary stage ascribed to the residual solvents and the adsorbed moisture inside the pores. PFcMOP began to be degenerated at 200 °C under nitrogen atmosphere, and the degeneration temperature of carbazole-based CFcMOP reached 400 °C, suggesting that the formation of a C-C covalent bond between the aldehyde group and benzene ring in carbazole was more stable than in pyrrole. At temperature from 200 to 800 °C, the continuous weight loss of the polymers was derived from the carbonization of MOPs attributed to the fact that TGA was carried out under the atmosphere of nitrogen. The X-ray diffraction test exhibited that they had amorphous morphology (Appendix A, ESI†). Like other microporous polymers, PFcMOP consisted of loose agglomerates of tiny particles with a rough surface and irregular shape, as investigated by SEM (Figure 1A,B). Interestingly, CFcMOP using carbazole as a connect unit can form sphere-like particles (Figure 1C,D).

The chemical connected structure of MOPs was determined by Fourier transform infrared spectroscopy (FTIR). As exhibited in the FTIR spectra (Figure 2), following the deformation (1680 cm^−1^) and strong reduction of the aldehyde in 1,1′-ferrocenedicarboxaldehyde, the adsorption bands at 2970 cm^−1^ were assigned to C_sp3_–H stretching vibrations indicative of the successful polymerizations reaction of substituent-group-free carbazole and pyrrole with 1,1′-ferrocenedicarboxaldehyde. In addition, the bands at around 1640 cm^−1^ are assigned to the stretching vibrations of N−H derived from carbazole and pyrrole rings. Furthermore, the elemental contents information in the as-prepared polymers was investigated by XPS spectra. As shown in Appendix A, the iron element signals were observed in the spectra, and the iron content values listed in Appendix A were 4.91% and 4.61% for PFcMOP and CFcMOP, respectively. In addition, the measured carbon content was 92.12% in CFcMOP, which agreed with the theoretical values (92.30%) and the higher carbon contents (91.30%) in PFcMOP than the calculated values (86.95%), which may be due to the absorption of CO_2_ in the porous polymers. Furthermore, the nitrogen content values were 3.17% and 3.24% for PFcMOP and CFcMOP, respectively. The difference between the measured value and theoretical value could be ascribed to the XPS test on the surface content of the polymers.

The porosity of the resultant ferrocene-based polymer networks was evaluated by the N_2_ adsorption and desorption isotherms at 77 K, as illustrated in Figure 3a. Both the two polymer networks exhibited a continual increase of N_2_ uptake at the relative pressure (P/P_0_) up to 1.0, suggesting the characteristics of mesopores. In addition, CFcMOP showed a steep rise at very low relative pressure (P/P_0_ < 0.01), and the N_2_ adsorption of CFcMOP increased slowly at relative pressure (P/P_0_) from 0.1 to 0.9, which can be roughly classified as type I sorption [41]. When the relative pressure exceeds 0.9, nitrogen uptake rapidly increases, suggesting the presence of meso- and macroporous structures probably derived from the interparticulate voids because of the loose packing of tiny particles, as detected by the field-emission SEM micrographs. Utilizing nitrogen as a probe, the pore sizes and distributions for the two microporous polymers were calculated through the non-local functional theory (NLDFT) from the adsorption isotherms of nitrogen at 77 K. As shown in Figure 3b, the PFcMOP exhibits the pore size of about 1.86 nm, and the CFcMOP shows a slightly smaller pore size located at 1.36 nm, which belongs to micropore region. Considering the higher molecular size of pyrrole than carbazole, the smaller pore size of carbazole-based CFcMOP than PFcMOP can be attributed to interpenetration effect, which is detected in many porous organic polymers [42,43].

The specific surface areas and porosity parameters of PFcMOP and CFcMOP were calculated from the adsorption isotherms. Using the Brunauer–Emmett–Teller (BET) model, the CFcMOP possesses specific surface areas up to 104.5 m^2^ g^−1^. However, the BET surface areas of PFcMOP using pyrrole as a building unit descended to 48.8 m^2^ g^−1^. That the BET surface areas of CFcMOP are a little higher than PFcMOP probably can be ascribed to the condensation reaction of the aldehyde group with two carbon atoms at the same five-membered ring of pyrrole, while there are different benzene units in carbazole, which affect the reacting activity of monomers. In addition, the pore volume of PFcMOP and CFcMOP reached 0.07 and 0.1 cm^3^ g^−1^, respectively.

Photocatalytic degradation of dyes was usually seen as an efficient and simple approach for the removal of dye pollutants due to its high efficiency, low cost and simplicity. Considering the existence of the metal ion in the microporous polymer networks, the photocatalytic activity of MOPs in methylene blue degradation under LED light irradiation was explored. Typically, the time-dependent adsorption of MB on 3 mg PFcMOP and CFcMOP was investigated at an initial concentration of 10 mg/L in the test tube with a magnetic stir bar. Firstly, the system was stirred without light for 60 min to keep the adsorption/desorption balance. After stirring at certain time intervals during the reaction processes, the absorbance variation of MB was detected at 660 nm by means of UV-vis spectroscopy in Figure 4. As seen in Figure 5, the efficiency of degrading MB could achieve 54% and 65% within 20 min for PFcMOP and CFcMOP, respectively. After stirring for 100 min, the degradation finally reached 65% and 78% for PFcMOP and CFcMOP, respectively. The higher photocatalytic activity of CFcMOP than PFcMOP can be ascribed to the larger special surface area of CFcMOP, in which MB molecules can easy diffuse into the pore channel of CFcMOP.

The photocatalytic active mechanism of MOPs was studied below. As is well known, four species including electron (e^−^), active hole (h^+^), hydroxyl radical (OH) and singlet oxygen (^1^O_2_) are presumed to react as active sites for photocatalytically degrading the dye [44]. Different scavengers, namely CuSO_4_, EDTA-2Na, tert-butanol (TBA) and L-His were selectively added to the photocatalytic reaction for quenching electron (e^−^), active hole (h^+^), ·OH and ^1^O_2_, respectively. As exhibited in Figure 6a, the presence of CuSO_4_, EDTA-2Na, L-His and tert-butanol led to a prominent restraint in the degradation of MB. It reflected that e^−^, h^+^, ^1^O_2_ and ·OH were reactive species in this degradation reaction. As shown in Figure 6a, the efficiency of PFcMOP degeneration reaction descended to 25%, 25%, 20% and 17%, respectively. In addition, the efficiency of CFcMOP degeneration reaction descended to 38%, 22%, 34% and 20% after adding corresponding scavengers to the degradation system (Figure 6b). Hence, in the MB photodecomposition reaction catalyzed by the MOPs, singlet oxygen and hydroxyl radical could be generated simultaneously as reactive species in the degradation of MB dye.

## 3. Conclusions

In summary, two ferrocenyl microporous organic polymers were successfully synthesized utilizing 1,1′-ferrocenedicarboxaldehyde as building monomers. The resultant polymers networks were examined by FTIR, XPS and SEM in detail. In addition, MOPs showed good thermal stability and possessed high porosity with a BET surface area from 48 to 105 m^2^ g^−1^, and MOPs could be used in degrading dyes with high efficiency.

## Data Availability

Not applicable.

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
