# Peer review of "Facile Synthesis of Microporous Ferrocenyl Polymers Photocatalyst for Degradation of Cationic Dye"

_polymers, 2022, doi:10.3390/polym14091900_

Round 1
Reviewer 1 Report
polymers-1653606-peer-review-v1
The current manuscript entitled “Facile synthesis of microporous ferrocenyl polymers photocatalyst for degradation of cationic dye” by “Zhang et al” deliberated on the Microporous organic polymers photocatalysts for degradation of dyes. The short communication manuscript seems good. But can be accepted after addressing the following comments.
- In the abstract starting “(MOPs)” use this abbreviation for Microporous organic polymers.
- In the abstract section mention the names of the synthesized polymers. Seems abstract is very less.
- Provide the full form of BET in the abstract section.
- Provide the full forms of polymers in the Figure 1.
- Label the functional groups for the peaks in the Figure 2.
- Grammatical errors are there at some places, please rectify.
- Provide some more information on the “type I sorption.”
- Please revise these lines, seems something is missing.
“Utilizing nitrogen as probe, the comparisons of pore
sizes and distributions for the two polymer networks were calculated by the non-local
functional theory (NLDFT) from the adsorption isotherms of nitrogen at 77 K.”
- Add some more information related to BET surface area. Detailed explanation is missing.
- Cite some potential literature on toxic pollutants.
https://doi.org/10.1016/j.ccr.2021.214305
Polymers 2022, 14(5), 880; https://doi.org/10.3390/polym14050880
Author Response
Dear Editor and reviewers,
I am writing this letter to thank editors and reviewers endeavour for reviewing the manuscript Polymers-1653606 titled “Facile synthesis of microporous ferrocenyl polymers photocatalyst for degradation of cationic dye”. There is obvious improvement of the manuscript after editors and reviewers’ scientific comments.
The revised manuscript was carefully corrected according to the referees' comments point by point. The revised detail as follows:
- In the abstract starting that “(MOPs)” usingabbreviation for Microporous organic polymerswas added.
- In the abstract section the names of the synthesized polymerswere mentioned and abstract was amended.
- The full form of BET in the abstract sectionwas provided.
- The full forms of polymers in the Figure 1was provided.
- The functional groups for the peaks in the Figure 2was labeled .
- Grammatical errors wererectified.
- More information on the “type I sorption.” was added in the manuscript“In addition, CFcMOP shows a steep rise at very low relative pressure (P/P0 < 0.01), and the N2 uptake of CFcMOP increases slowly at relative pressure(P/P0) from 0.1 to 0.9, which can be roughly classified as type I sorption.”, and the similar N2 uptake isotherms were observed in the references, such as “Moon, H. R., Kobayashi, N., & Suh, M. P. (2006). Porous Metal−Organic Framework with Coordinatively Unsaturated MnII Sites: Sorption Properties for Various Gases. Inorganic chemistry, 45(21), 8672-8676.” and
“Yu, H., Tian, M., Shen, C., & Wang, Z. (2013). Facile preparation of porous polybenzimidazole networks and adsorption behavior of CO2 gas, organic and water vapors. Polymer Chemistry, 4(4), 961-968.”
- The sentence that ““Utilizing nitrogen as probe, the comparisons of pore
sizes and distributions for the two polymer networks were calculated by the non-local
functional theory (NLDFT) from the adsorption isotherms of nitrogen at 77 K.”” have been modified as “Utilizing nitrogen as probe, the pore sizes and distributions for the two polymer networks were calculated by the non-local functional theory (NLDFT) from the adsorption isotherms of nitrogen at 77 K.”
- Explanation that the BET surface areas of CFcMOPexhibits little higher than PFcMOPwas depicted
- 10. Relatedliteratureson toxic pollutants of “https://doi.org/10.1016/j.ccr.2021.214305” and “Polymers 2022, 14(5), 880; https://doi.org/10.3390/polym14050880” were cited, seeing literature 23 and 26.
We would feel honored if the paper could be accepted. Thank you again for your endeavour toward to reviewing the manuscript.
Yours Sincerely
Gen Li
Qingquan Liu
Reviewer 2 Report
The manuscript reports preparation of a new microporous organic polymer (MOP) by polycondensation of carbazole and pyrrole with ferrocenedicarboxaldehyde. The photocatalytic activity of prepared MOP is demonstrated in methylene blue (MB) degradation. The manuscript is worth publishing, however, after some revision.
- As concerns elemental analysis of polymer: the determination of carbon content from XPS is usually not reliable because of absorption of CO2 and other C containing impurities, the Fe/N ratio seems to me more instructive. XPS is a surface method; would it be possible to determine Fe content by elemental analysis in bulk?
- The detailed condition of photocatalytic experiments should be given: solvent, temperature, source of illumination, speed of stirring, etc. (e.g. as a paragraph in ESI).
Similarly the measurements of UV- vis spectra: apparatus, lengths of cuvette, scale on absorbance axis are missing.
- The efficiency in MB degrading was higher for CFcMOP than for PFcMOP and the difference was ascribed to the different surface area. However, from Fig 5 it seems that catalytic centres in PFcMOP lose their activity more quickly than centres in CFcMOP. Could you comment it?
- The effect of scavengers on MB degradation needs better discussion: what does it mean “the efficiency of….reaction descend to 25%, ….?, why second part of Fig.6 is not mention in discussion?, the reaction condition (especially the scavenger concentrations) should be added.
- Please, check the reference 6. 22 should be probably 27.
Author Response
Dear Editor and reviewers,
I am writing this letter to thank editors and reviewers endeavour for reviewing the manuscript Polymers-1653606 titled “Facile synthesis of microporous ferrocenyl polymers photocatalyst for degradation of cationic dye”. There is obvious improvement of the manuscript after editors and reviewers’ scientific comments.
The revised manuscript was carefully corrected according to the referees' comments point by point. The revised detail as follows:
1 Due to the hyper-cross-linked nature of MOPs, elemental analysis was sometimes not accurate, for examples, the results were found in the literatures such as “ACS Catal. 2021, 11, 1008−1013
”, “Macromolecules 2017, 50, 4993−5003”, “ACS Appl. Mater. Interfaces 2020, 12, 47984−47992”, “ACS Appl. Mater. Interfaces 2016, 8, 32060−32067”, “Chem. Commun., 2014,50(16), 1937-1940.”,“J. Mater. Chem. A, 2014, 2(21), 7795-7801.” et al. Hence, in this paper, PFcMOP and CFcMOP were analyzed by XPS to give the description for reference.
2 The detailed condition of photocatalytic experiments was given in the supporting materials,and the measurements of UV- vis spectra was also added.
3 According to Fig. 5, at initial stage, catalytic centres in PFcMOP lose their activity more quickly than centres in CFcMOP. With the reaction increasing, the higher surface area of CFcMOP showed higher MB degrading efficiency than PFcMOP.
4 The second part of Fig.6 was discussed, and the reaction condition was described in supporting information.
5 The reference 6. 22 was rectified as 27.
We would feel honored if the paper could be accepted. Thank you again for your endeavour toward to reviewing the manuscript.
Yours Sincerely
Gen Li
Qingquan Liu
Round 2
Reviewer 2 Report
The elemental analysis of new polymers is essential. As this analysis is based on XPS only, the more detailed discussion of XPS spectra is necessary. Please add the assignment of the individual peaks and discuss the difference between XPS spectra of both polymers on Fig. S3. I suppose that the signal of Fe is at about 710 eV; why in spectrum of CFcMOP the intensity of this signal is so low.
Author Response
Dear Editor and reviewers,
I am writing this letter to thank editors and reviewers endeavour for reviewing the manuscript Polymers-1653606 titled “Facile synthesis of microporous ferrocenyl polymers photocatalyst for degradation of cationic dye”. There is obvious improvement of the manuscript after editors and reviewers’ scientific comments.
The revised detail as follows:
Signals of C and N elements in XPS spectra were assigned, and XPS spectrum of Fe element about 710 eV in the polymers was given in the Supporting Information. The C,N, Fe elements content of the polymers were calculated, and listed in Table S1.
That the experimental value was not consistent with theoretical value was also found in the literatures such as “ACS Catal. 2021, 11, 1008−1013 ”, “Macromolecules 2017, 50, 4993−5003”, “ACS Appl. Mater. Interfaces 2020, 12, 47984−47992”, “ACS Appl. Mater. Interfaces 2016, 8, 32060−32067”, “Chem. Commun., 2014,50(16), 1937-1940.”,“J. Mater. Chem. A, 2014, 2(21), 7795-7801.” et al. The element contents of PFcMOP and CFcMOP was analyzed by XPS to give the description for reference.
We would feel honored if the paper could be accepted. Thank you again for your endeavour toward to reviewing the manuscript.
Yours Sincerely
Gen Li
Qingquan Liu
Round 3
Reviewer 2 Report
- The authors improved the XPS spectra (Fig. S3 and Fig.S4) and added Table S1 with content of C, N, and Fe elements. It gives a better insight into product composition. In this respect, the main text (p.3) should be modified accordingly: (a) there is practically perfect agreement between theoretical and experimental values of C content, (b) however, in both polymers, the content of N is lower and the content of Fe is higher than the theoretical values. It may indicate that N-containing monomers reacted incompletely. Could you comment it?
- The polymer preparation is described in SI sufficiently; however, the polymer yields are missing. Could you add these data, please?
Author Response
Dear editors and reviewers,
I am writing this letter to thank editors and reviewers endeavour for reviewing the manuscript Polymers-1653606 titled “Facile synthesis of microporous ferrocenyl polymers photocatalyst for degradation of cationic dye”. There is obvious improvement of the manuscript after editors and reviewers’ scientific comments.
The revised detail as follows:
1, The main text on the element content was modified, and the the difference between the measured value and theoretical value was explained in the page 3.
2, The polymer yieldswere given in the supporting information.
We would feel honored if the paper could be accepted. Thank you again for your endeavour toward to reviewing the manuscript.
Yours Sincerely
Gen Li
Qingquan Liu